# Assessing the Emotional Affordance of Brand Image and Foreign Image Based on a Physiological Method Using Examples from Dubai: Exploratory Study

**Ala' Albdour** [1] , **Ahmed Agiel** [1,*] and **Kilani Ghoudi** [2]

1 Department of Architectural Engineering, College of Engineering, United Arab Emirates University, Al Ain 15551, United Arab Emirates
2 Department of Analytics in the Digital Era, College of Business and Economics, United Arab Emirates University, Al Ain 15551, United Arab Emirates
* Correspondence: a.agiel@uaeu.ac.ae

**Abstract:** The United Arab Emirates (UAE) is a country with few restrictions on architectural styles. The main aim of this paper was to investigate a group of participants' unconscious emotional responses to images of traditional and modern architectural styles in the UAE. All images were from the city of Dubai, but participants were from elsewhere to avoid the influence of familiarity. A physiological method was used to measure the unconscious emotional responses to the images' visual contexts, specifically the emotions of stress, engagement, interest, focus, excitement, and relaxation. Six architects working in the UAE were then interviewed for their interpretations of both the images themselves and the participants' emotional responses. A sample of 29 male laypeople (aged 18–45) participated in this study, divided into locals from Al Ain city, locals from the northern emirates, and nonlocals. The results showed that the brand image provided observers with better emotional quality than the foreign image for local participants from Al Ain, and that nonlocal participants showed strong emotional responses to the traditional architecture, but the northern Emiratis remained neutral. The findings of this study contribute to emotion studies in the field of meaning in architecture. It also validates the effectiveness of a physiological method of investigating the emotional responses to architectural styles.

**Keywords:** contemporary architecture; architectural styles; emotional quality; EMOTIV; EPOC+; EEG

## 1. Introduction

Design in the urban context has been one of the main debates between world-class architects in the last seven decades. Architects including Rem Kollhass, Zaha Hadid, Frank Gehry, and Daniel Libeskind, inspired by Jacques Derrida's deconstruction theory, denounced the concept of respecting the context in designing new buildings, at the same time that architects such as Tadao Ando, Peter Zumther, Renzo Piano, and Michael Graves were desperately calling for respecting the context [1]. On the one hand, the debate between architects has not yet concluded, and it is usually based on endless theoretical arguments that lack scientific evidence. On the other hand, science-based studies would provide helpful evidence to better guide the theoretical debates about context and cultural sustainability. Additionally, those studies would also contribute to the debate about cultural sustainability [2], which is considered a missing fourth pillar of sustainable development [3].

In the last decade, increasing numbers of researchers have explored the relationships between built environments and human emotions, perceptions, and cognitions. In this field, at least five subfields are directly related to environmental perception and architectural styles: place attachment, psychological restoration, scientific aesthetics, neuroarchitecture, and environmental psychology. Place attachment is typically concerned with the relations between meaningful places and people [4,5]. This relationship can be described as a bond

between people and areas generated across time [6], where those people feel safer, more comfortable and prefer to stay there [7]. Psychological restoration was developed based on the attention restorative theory (ART) of Kaplan and Kaplan, which proposes the capability of the natural environment to enhance the concentration of people [8]. Later on, researchers in this field started to investigate the restorative capability of built environments [9,10], or to compare them with the natural environment [11,12]. Scientific aesthetics, meanwhile, is a very interesting field that is concerned with exploring the cognitive processes and feelings associated with aesthetic appeal [13–15]. A well-established branch in this field is concerned specifically with environmental aesthetic [16–18]. Neuroarchitecture is a relatively new field that emerged 22 years ago, concerned with combining the field of neuroscience with the field of architecture [19]. Scholars in this field have explored the neural impact of different geometries and properties of spaces [20], the density of curvature or linearity of interior spaces [21], and even architectural styles, investigated by Choo et al. [22], who found differing neural activity in the high-level visual region of the brain associated with different architectural styles. There are also other scholars concerned with the built environment and human cognition, but in terms of wayfinding, such as Hölscher et al. [23,24] and Wiener et al. [25], or even in terms of designing spaces for people with no memory, such as Alzheimer's patients [26]. However, studies about the built environment and human emotions, perception and cognition are interdisciplinary.

This study was conducted to contribute to the field of neuroarchitecture based on previous studies and dissertations in the field of environmental psychology that have specifically studied the emotional qualities of laypeople who are exposed to traditional and modern visual contexts. Examples of these studies include those by Agiel [27] and Agiel, Lang, and Caputi [28], who explored the existence of emotional connections between visual contexts and people, as well as the impacts of those contexts on people's attachment to the place. Mastandrea, Bartoli, and Carrus [29] found that the stimuli of classic architecture were quickly processed and were connected with positive words without difficulty, in contrast with the stimuli of contemporary architecture. Stamps and Nasar [30] also reported that popular styles were preferred over avant-garde styles, but Stamps [31] reported the opposite. Chokor and Stamps [32,33] concluded that people preferred modern buildings to popular buildings, but Groat [34], Devlin [35], Nasar [34–36], and Purcell and Nasar [36] concluded the opposite. These researchers explored laypeople's emotional connections and preferences based on their conscious feedback on self-report assessments, with items rated on a Likert scale, except for a dissertation that was based on participants' conscious and subconscious interpretations based on personal construct theory and its analytical components. The results of studies that rely on conscious interpretations have been inconsistent. Within the present study, which is a part of a master's thesis, we explore participants' unconscious interpretations. Unconscious interpretations are data that are collected without participant interference based on measuring participants' physiological attributes. Studies that explored the unconscious responses of participants on architectural styles are very limited. The research question in this study is whether architectural styles that relate to the past afford better emotional qualities than the ones that do not, and our hypothesis is that they do. This study contributes to the knowledge on meaning in architecture and on understanding the gap between theory and practice. Moreover, it provides new scientific evidence to enhance the architectural debate on theory and practice, and consequently the debate about the missing cultural sustainability pillar.

## 1.1. Theoretical Background

Many authors have written about architectural styles and movements in general. For our study, we reviewed three recent books and one doctoral dissertation examining this issue to explore the most current points of view [27,37–39]. Each of these authors uses a different paradigm to classify architectural styles (see Table 1). In the paradigm created by Agiel [27], buildings can be classified under one of five factors: brand image, local image, foreign image, new image, and ideal image. These factors were introduced

based on the understanding of people's interpretation of the appearance of buildings [27]. Lang and Moleski [37] categorize architectural styles into neo-modernist, postmodernist, revivalist and neo-traditional, critical regionalist, and ecological architecture. Hopkins [38] developed a similar classification of architectural styles that encompassed regionalism, eco-architecture, expressive rationalism, and contextualism. Haddad and Rifkind [39] categorize architectural styles between 1960 and 2010 into two major pillars: (1) architectural developments since modernism and (2) architectural developments around the world. The first pillar encompasses modern architecture circa 1959, postmodernist, high tech, deconstructionist, greening, and other postcolonial theories of architecture [39]. The paradigms introduced by Lang and Moleski [37] and Haddad and Rifkind [39] attempted to categorize contemporary architecture while considering two aspects: the formalistic aspect and the ideological aspect. Because both paradigms included ideological aspects, the architectural styles in the two paradigms are not easily distinguishable by laypeople. The postmodernism movements, for example, reflect the ideology of pluralism rather than singularity; that is, they share a range of overlapping interests.

**Table 1.** Comparison between architectural style paradigms.

| | Architectural Style Paradigms | | | |
|---|---|---|---|---|
| | **Lang and Moleski [37]** | **Haddad and Rifkind [39]** | **Hopkins [38]** | **Agiel [27]** |
| Architectural style categories | Neo-modernist (1- architecture of structural and geometrical dexterity, 2- architecture of abstract signification). Postmodernist (1- architecture of postmodern exuberance, 2- deconstructionism). Revivalist and neo-traditional. Critical regionalist. Ecological. | Modern architecture circa 1959. Postmodernist. High-tech. Deconstructionist. Greening architecture. Postcolonial theories in architecture. Architectural developments around the world. | Regionalism. Eco-architecture. Expressive rationalism. Contextualism. | Brand image. Local image. Foreign image. New image. Ideal image. |
| Criteria | Architecture ideology | Architecture ideology | Mix between ideology and form | Visual perception |
| Distinguishability by laypeople | Difficult | Difficult | Possible (except eco-architecture) | Possible |
| Special features | N/A | N/A | N/A | This classification pertains to perceived image, not just the architectural style |
| Suitability for this study | Not suitable | Not suitable | Suitable | Ideal |

Hopkins's paradigm [38] is more straightforward than those of Lang and Moleski [37] and Haddad and Rifkind [39] in terms of distinguishability by laypeople. However, there remains a gap in the ability to distinguish eco-architecture from other styles. For instance, in Hopkins's paradigm [38], it can be difficult in some cases to judge whether a building is a type of eco-architecture or another style simply from its form, and this confusion is based on the similarities of the characteristics between eco-architecture and other styles. This dilemma led Agiel [27] to construct a new straightforward classification of building forms, in which he based classification on a psychological experiment conducted among laypeople and architects that led him to introduce five factors to describe current existing buildings. The classification criteria in Agiel's paradigm are consistent with the purposes of this research.

In Agiel's paradigm, the brand image represents traditional architecture, and the local image represents neo-traditional architecture, which is the product of reworking and redesigning the brand image. The foreign image and new image both represent global architecture; whereas the foreign image represents the global architecture the locals are

familiar with through long visual contact, the new image represents unique global trends in architecture.

The two images we selected for examination in this study were the brand image as a representation of traditional visual context and the foreign image as a representation of international visual context.

## 2. Data Collection and Methods

We conducted this explorative research using quantitative methodology to explore laypeople's emotions and brain performance in response to a specific attribute within a built environment. We also used a qualitative method, semi-structured interviews, to gain more detailed insights into the results of the empirical study. In the following section, we describe the quantitative methodology and subsequently the qualitative methodology.

### 2.1. Quantitative Methodology and Experiment Details

Nasar [40] cited four components necessary for experiments in assessing built environments: (1) selecting and measuring the environmental attributes, here including their impacts on the participants; (2) selecting the relevant environmental stimuli and determining how best to present those stimuli to participants; (3) selecting the measurement tools to capture the effects of interest, here emotional impacts of architectural styles; and (4) selecting respondents. Lastly, we added a fifth component to be addressed in this study, which is (5) the affected emotions: the emotions that are affected by changes in environmental attributes. We address these five components below before proceeding to the experiment.

#### 2.1.1. Selecting and Measuring Environmental Attributes

Historical significance is one of the six physical attributes identified in research that people highlight in their perceptions of the environment, along with naturalness, upkeep, openness, complexity, and order [40]. Nasar [40] suggests selecting one of these environmental attributes and measuring their impacts on participants' emotions, defining them as follows: *naturalness* refers to the predominance of nature elements such as vegetation, water, and mountains; *upkeep* refers to perceived maintenance; *openness* refers to the distance between buildings and visual scope; *complexity* refers to the number of buildings and the complexity of structures; *order* refers to or indicates unity, consistency, and legibility; and *historical significance* refers to historical authenticity according to the observer and to resembling a historical environment. For this study, we selected two contradictory environments in terms of their historical significance: (1) an environment with a brand image that looked historical to the participants and (2) an environment with a foreign image that did not look historical to the participants.

#### 2.1.2. Environmental Stimuli and Stimulation Method

The mode of presentation or stimulation method is an important aspect of environmental stimuli. Typical methods of presenting experimental conditions to participants are either through direct perception or observation or through mediated perception using retinal, neural, or mental pictures [41]. Gibson [41] defines direct perception as "the activity of getting information from the ambient array of light a process of information pickup that involves the exploratory activity of looking around, getting around, and looking at things" (p. 139).

In this study, direct perception could have involved a walk-through, and mediated perception could have taken place through observing colored or black and white still or video images. However, both methods have limitations: walk-throughs can involve numerous external influences on participants' emotions, including the presence of other people, sound, wind speed, temperature, transportation of participants, and participant exhaustion; and mediated perception (videos and photographs) cannot provide the third dimension that Gibson discusses: the exploratory activities of moving around and observ-

ing things. Therefore, we selected virtual reality using third-party 360-degree photographs of the selected environments as the most realistic and suitable technology. The use of this technology avoided most of the limitations of direct and mediated perception [42]. The photographs were selected from more than a thousand 360-degree photographs taken by a third party (Dubai Film), and we selected the images based on the historic value of their environments as a main variable.

Al Fahidi Historical District (also known as Al Bastakiya) is an outdoor shopping district located in Dubai. This district is unique not only to Dubai but also to the United Arab Emirates (UAE), because there is no other outdoor built environment with an equivalent historical value that has not been overtaken by modern developments [43] (pp. 179–180). The area is also unique because it contains all the elements of the UAE's coastal traditional architecture; therefore, it reflects the coastal brand image of the UAE. Dubai contains numerous modern built environments, and a group of 12 architects at different firms in the UAE selected Al Bastakiya for its physical attributes (complexity, order, openness, upkeep, and naturalness) from among five alternatives. Figure 1 shows the five locations.

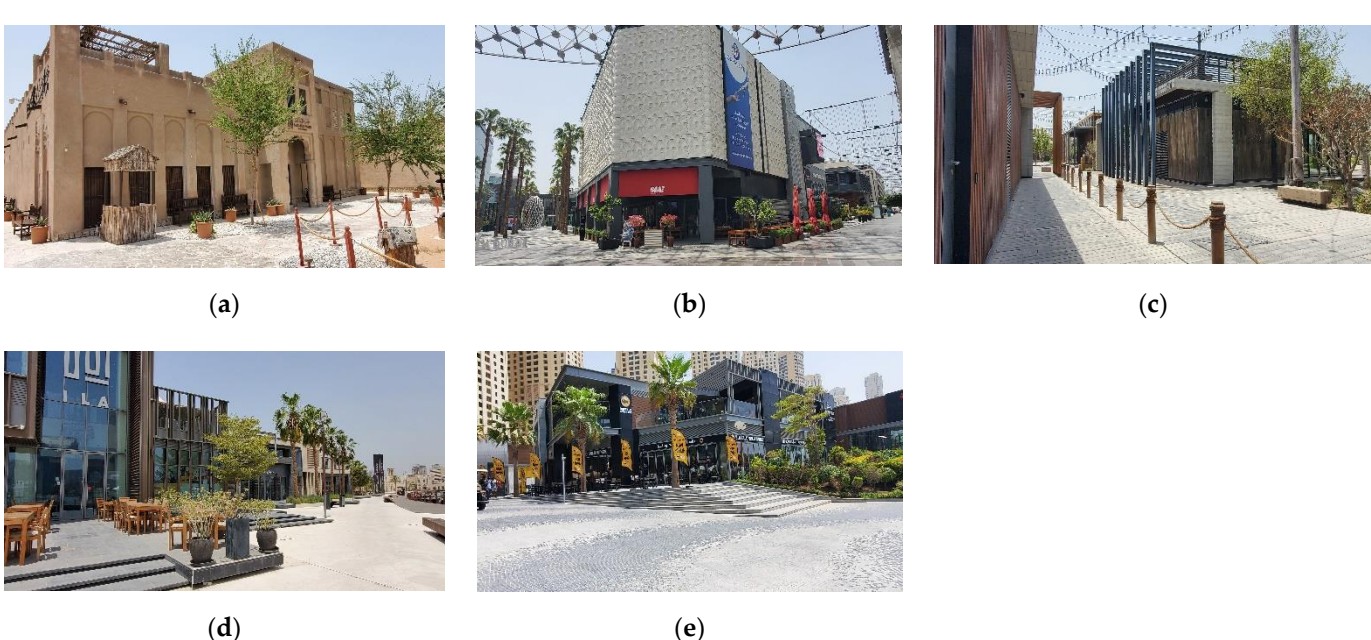

**Figure 1.** Historical and modern built environments in Dubai. All images taken by author, 2022. (**a**) Al Fahidi Historical District (Al Bastakiya), reconstructed in 1971; (**b**) City Walk, constructed in 2016; (**c**) La Mer, constructed in 2017; (**d**) Al Seef, constructed in 2017; and (**e**) The Walk (Jumeirah Beach Residence), constructed in 2007.

The 12 architects were invited to evaluate the salient attributes of each location in terms of their similarities, and the group ranked Al Bastakiya and City Walk as matching in terms of their physical attributes (ranked as low, medium, or high). Table 2 shows the comparison of architects' evaluations of the foreign visual contexts with the brand visual contexts. Table 2 reflects the average evaluations of the 12 architects on each of the six attributes we defined earlier. We used this comparison to select the foreign built environment as the stimulus in this research.

**Table 2.** Comparison between built environments' physical attributes.

| Place | Complexity | Order | Openness | Upkeep | Naturalness |
|---|---|---|---|---|---|
| Al Bastakiya | High | High | medium | High | Low |
| City Walk | High | High | medium | High | Low |
| La Mer | High | High | High | High | High |
| Al Seef | High | High | High | High | High |
| The Walk | High | High | High | High | High |

2.1.3. Response Measures (Measurement Tool)

Researchers reported many tools for measuring emotions, usually categorized within three approaches: (1) self-reporting or verbal measures (e.g., the Self-Assessment Manikin, questionnaires, or interviews); (2) behavioral analysis (e.g., facial expressions, such as the Facial Action Coding System or facial electromyography); and (3) physiological methods (e.g., cardiac rhythm, physiological signals, or respiration) [40,44]. Previous researchers have used a physiological method, specifically EEG signal analysis, in studies on emotions and the built environment [11,45] and in multiple studies to detect human emotions unconsciously and to test its credibility [44,46–50]. However, none of those researchers investigated the impact of the historical significance of the built environment on participants' emotions. In contrary to the functional magnetic resonance imaging (fMRI) method which was used by some researchers [22,51] and which requires the participant to be inside the machine in the lab [52], the EEG method is used more often because of the availability of portable low-cost EEG devices that can provide researchers with EEG data similar to that provided by clinical devices used in the hospitals [22,50,51]. Therefore, we used an EPOC+ device from Emotiv to provide us with EEG signals from the participants.

2.1.4. Respondents

Respondents in this type of study are classified as residents, passersby, or occasional visitors [40]. In the current study, the environmental attribute under examination was historical significance; therefore, the respondents had to be aware of the environment's historical significance to test the hypothesis that traditional architectural styles elicit more positive emotions. To ensure this, it was essential that participants were living in the UAE. Moreover, the factor of the affective bond that could be created between people and specific regions had to be considered. In other words, if an architectural structure is found in a certain place that people associate with positive feelings, people might be inclined to provide positive feedback concerning the structure [7]. Therefore, to avoid any bias that could arise from familiarity and memories of the place [40], participants needed to not be living close to the selected environment. Meanwhile, historical value is argued to be relative to one's place of living, and public preferences are affected by the geographic location of the participants [35,53–55]. To avoid the familiarity effect and the geographic location impact, participants were selected from Al Ain city, which is 120 km from Dubai. Ethical approval (number ERS_2019_5878) for the experiment was obtained from United Arab Emirates University Research Ethics Committees. All the participants were male because of a limitation in the method which was used to collect the data, mentioned in the limitations section, in addition to the nature of the UAE population, which contains 275 males for every 100 females for all residents aged 15–49 according to the United Nations' 2020 statistics [56]. The age group of the participants (18–45) represented 67.6% of the UAE population, which has a median age of 32.62 [56]. We selected three groups of participants based on where they were from: (1) locals from Al Ain city, selected using the snowballing method ($n = 7$); (2) locals from the Northern Emirates who were living and studying in Al Ain city, who responded to emails sent to UAEU students living in the university dormitories and to advertisements posted in the dormitory buildings ($n = 12$); and (3) nonlocal Arabs from Jordan and Egypt who were living in Al Ain city, selected using the snowballing method ($n = 10$). The first participants from groups (1) and (3) were recruited with the help of two participants from group (2). The demographic data of the participants are shown in Table 3.

**Table 3.** Participants' demographics.

| Nationality | No. of Participants | Age Group (18–30) | Age Group (30–45) | Abandoned Participants | Total |
|---|---|---|---|---|---|
| Locals: Al Ain | 7 | 5 | 2 | N/A | 7 |
| Locals: Northern Emirates | 13 | 13 | | 1 | 12 |
| Nonlocals | 13 | 10 | 3 | 2 | 10 |

### 2.1.5. Affected Emotions

In his book The Expression of the Emotions in Man and Animals, Charles Darwin wrote about his study of emotions, which was based on the observation of the facial and bodily expressions of animals and infants [57]. Several decades after Darwin, the psychologist Paul Ekman, widely known for his theory of basic emotions, claimed that all emotions were derived from six basic emotions: happiness, surprise, fear, anger, disgust, and sadness [58].

Roberts [59] argues that classifying basic emotions is important for producing a set of generalizable emotions for all people. Ortony and Turner [60] attempted to track the emergence of this idea over the past two centuries. They argue that the concept of basic emotions is not plausible.

In contrast, another theory of emotions defines emotions under two dimensions, valence and arousal; others argue that dominance is a third dimension. The idea of describing emotional status with two dimensions was initiated by Wundt [61] (pp. 51–53), although he did not call them valence and arousal. Later, Schlosberg [62] attempted to classify emotions in terms of two dimensions based on facial expression: pleasantness–unpleasantness and attention–rejection, which today are known as valence (or pleasantness) and arousal. Mehrabian and Russell [63] suggested a three-dimensional approach to studying environmental psychology based on facial expressions; they argued that pleasure, arousal, and dominance are capable of describing the emotional status of observers. Mehrabian and Russell [63] interpreted pleasure as ranging between happiness and unhappiness, arousal as ranging between sleep and excitement, and dominance as a feeling of control and restriction of behavior ranging between dominance and submission. A new significant study has supported Ortony and Turner's argument, finding that knowledge about emotions is abstract and highly dimensional, and that other emotions are not reducible to primitive models, such as the six basic emotions or the valence/arousal model [51]. However, researchers have recently been divided between two approaches: the three-dimensional approach (i.e., arousal, valence, and dominance) and the multiple emotions approach.

This research adopts the multiple emotions approach to contribute to the accumulated knowledge on this topic by exploring how architectural styles induce new emotions that have not been investigated before. We also adopted multiple emotions because the EPOC+ headset, which records real-time EEG signals, provides an algorithm for translating EEG signals into six performance metrics indicating six emotions (excitement, interest, stress, engagement/boredom, focus, and relaxation). The company describes these emotions on the performance metrics page of its website [64].

### 2.1.6. Experiment

We had difficulty convincing people to participate because of the duration and nature of the experiment, which involved fixing an EEG device on participants' heads. This latter aspect also determined the sex of participants, who were all male, because the experiment required fixing sensors in the hair. In fact, we still found it difficult to affix the sensors with good connections when the participant had long, dense hair.

This study was conducted in a quiet room with participants seated on chairs. The room temperature was 24–26 °C, and lighting was moderate. Some experiments were abandoned because of unexpected noise or connectivity issues. Participants were exposed

to a short explanation about the experiment 5 min before the start of the experiment. The procedure involved fixing the device to the top of participants' heads and connecting it to the laptop, which took 5–10 min. After a sufficient signal was established, four scenes were randomly shown to participants on a smartphone-based virtual reality headset that converted a split window of a 360-degree photograph on a smartphone screen to a virtual reality environment. Each scene was comprised of three major angles, and participants were told to focus on the buildings in each angle for 10–13 s until the EPOC+ device provided readings (the EPOC+ device provides new readings every 7 s).

After each reading was obtained, participants were instructed to view the next angle. After finishing all angles in the scene, the participants were told to remove the VR headset for 2 min in order to prepare for the next scene. The smartphone screen was mirrored on the laptop screen so that the researcher was aware of the angle that the participants were observing. The entire procedure, from the participants' observations of the first scene to the last (45 min on average), was recorded by both audio and laptop screenshots or videos, and the participants showed no discomfort or dissatisfaction after finishing the experiments, perhaps because they had prepared for that duration or because of the 2 min breaks between each scene.

### 2.2. Qualitative Methodology

We conducted 30 min semi-structured interviews after we analyzed the results of the conducted experiment to enrich the discussion and reflect on the outcomes of the experiment in the context of architecture practice. Therefore, we conducted the interviews with six Emirati, Jordanian, and Egyptian architects working in the UAE. There are no special institutes or associations for architects in the UAE which can provide us with a database for architects in the UAE. Therefore, we used the convenience sampling method to interview 2 architects chosen based on their experience in the UAE and direct contact with clients, then we relied on the snowballing method to reach the remaining 4 architects (see Table 4). However, we formulated the interview questions based on the results from the experiment as follows:

- Could you tell us about your experience in the field of architecture in the UAE and your knowledge about traditional architecture in Alain and northern emirates?
- Is it possible to mention the theory of architecture behind designing the contexts shown on the screen?
- We have conducted an experiment with three groups of participants: locals from northern emirates, locals from Al-Ain, and Jordanian and Egyptian expats living in Alain. The results showed a type of emotional attachment between the traditional architecture of the UAE and both the second and third groups of participants. Based on your experience, why do you think they felt this way?

**Table 4.** Architect selection method.

| Arch. No. | Nationality | Experience in the UAE | Selection Method |
|:---:|:---:|:---:|:---:|
| Arch 1 | Jordan | 6 | Convenience |
| Arch 2 | Egypt | 5 | Snowballing |
| Arch 3 | Jordan | 9 | Convenience |
| Arch 4 | Jordan | 6 | Snowballing |
| Arch 5 | UAE | 15 | Snowballing |
| Arch 6 | UAE | 13 | Snowballing |

We conducted the interviews online using Zoom and recorded each one; at the beginning of each, we explained the purpose of the interview. We showed the interviewees the same photos we had shown to the participants to ask them about the theory of architecture behind the design concepts in the scenes. After the interviews, we conducted thematic

analysis of the interview transcripts and derived four themes: (1) the architects' experience in the UAE, (2) their knowledge of architectural styles and theories, (3) their impressions of the architectural styles shown in the scene, and (4) their interpretations of our study results.

## 3. Data Analysis and Results

We used the Wilcoxon signed-rank test to objectively analyze the collected data from the empirical part of the study to validate the following research hypothesis: *The architectural styles that relate to the past afford better emotional quality than do architectural styles that do not relate to the past.* The Wilcoxon signed-rank test is a nonparametric—that is, it does not assume the normality of the data—analysis method that does account for the magnitude of the observations [65]. We also used a paired *t*-test including bootstrapping to check the data for robustness. We used these statistical methods because of the nature of the collected data from the EPOC+ device, and because validation requires a highly credible quantitative method.

We conducted three sets of analyses on the data collected from each group of participants, reflected in three boxplot graphs for each group (presented in the next section). For each set of analyses, we performed the Wilcoxon signed-rank test and the paired *t*-test with bootstrapping. In the first set, we compared the EEG results from the daytime brand image scene with the readings from the foreign image scenes. In the second analysis, we compared the EEG results of the nighttime brand image scene with the readings from the foreign image scenes. In the third analysis, we compared the EEG results of the daytime brand image scene with the readings from the nighttime brand image scenes. The two statistical tests gave concordant results for all comparisons. Table 5 gives the Wilcoxon signed-rank test findings, and Table 6 presents the results for the paired *t*-tests with bootstrapping.

**Table 5.** Summary of the Wilcoxon signed-rank test results for all participants.

| Scene Comparison | Emotion | Nationality [a] | | | | | |
| --- | --- | --- | --- | --- | --- | --- | --- |
| | | Al Ain Locals | | Nonlocals | | Northern Emirates Locals | |
| | | Z | Asymp. Sig. (2-Tailed) | Z | Asymp. Sig. (2-Tailed) | Z | Asymp. Sig. (2-Tailed) |
| Traditional daytime—Modern nighttime | Stress | −1.014 [b] | 0.310 | −1.420 [b] | 0.156 | −0.978 [b] | 0.328 |
| | Engagement | −0.507 [b] | 0.612 | −0.031 [b] | 0.975 | −0.267 [b] | 0.790 |
| | Interest | −0.338 [c] | 0.735 | −0.738 [b] | 0.460 | 0.000 [d] | 1.000 |
| | Focus | −2.197 [b] | 0.028 | −2.500 [b] | 0.012 | −0.314 [c] | 0.754 |
| | Excitement | −2.028 [b] | 0.043 | −1.533 [b] | 0.125 | −1.059 [c] | 0.289 |
| | Relaxation | −0.085 [b] | 0.933 | −0.284 [b] | 0.776 | −1.609 [b] | 0.108 |
| Traditional nighttime—Modern nighttime | Stress | −0.676 [b] | 0.499 | −1.136 [b] | 0.256 | −1.334 [b] | 0.182 |
| | Engagement | 0.000 [d] | 1.000 | −0.251 [b] | 0.802 | −0.800 [c] | 0.424 |
| | Interest | 0.000 [d] | 1.000 | −2.613 [b] | 0.009 | −0.471 [c] | 0.638 |
| | Focus | −0.507 [b] | 0.612 | −0.284 [c] | 0.776 | −0.471 [b] | 0.638 |
| | Excitement | −2.366 [b] | 0.018 | −0.966 [b] | 0.334 | −0.157 [b] | 0.875 |
| | Relaxation | −0.169 [b] | 0.866 | −1.988 [b] | 0.047 | −0.235 [b] | 0.814 |
| Traditional daytime—Traditional nighttime | Stress | −0.676 [b] | 0.499 | −0.063 [c] | 0.950 | −0.039 [b] | 0.969 |
| | Engagement | −0.338 [b] | 0.735 | −0.353 [c] | 0.724 | −1.426 [b] | 0.154 |
| | Interest | −0.169 [b] | 0.866 | −1.449 [c] | 0.147 | −0.275 [c] | 0.784 |
| | Focus | −1.352 [b] | 0.176 | −1.733 [b] | 0.083 | −1.172 [c] | 0.241 |
| | Excitement | −1.859 [b] | 0.063 | −0.795 [b] | 0.427 | −1.020 [c] | 0.308 |
| | Relaxation | −0.338 [c] | 0.735 | −1.903 [c] | 0.057 | −0.889 [b] | 0.374 |

Note. [a] Wilcoxon signed-rank test; [b] based on negative ranks; [c] based on positive ranks; [d] sum of negative ranks = sum of positive.

**Table 6.** Summary of the bootstrap results for the paired-samples *t*-tests for all participants.

| Scene Comparison | Emotion | Nationality | | |
| --- | --- | --- | --- | --- |
| | | Al Ain Locals | Nonlocals | Northern Emirates Locals |
| | | Sig. (2-Tailed) | Sig. (2-Tailed) | Sig. (2-Tailed) |
| Traditional daytime—Modern nighttime | Stress | 0.372 | 0.194 | 0.388 |
| | Engagement | 0.553 | 0.893 | 0.705 |
| | Interest | 0.677 | 0.218 | 0.712 |
| | Focus | 0.026 * | 0.041 * | 0.761 |
| | Excitement | 0.037 * | 0.125 | 0.231 |
| | Relaxation | 0.837 | 0.572 | 0.115 |
| Traditional nighttime—Modern nighttime | Stress | 0.389 | 0.203 | 0.260 |
| | Engagement | 0.868 | 0.828 | 0.419 |
| | Interest | 0.738 | 0.022 * | 0.887 |
| | Focus | 0.606 | 0.799 | 0.772 |
| | Excitement | 0.102 | 0.508 | 0.921 |
| | Relaxation | 0.992 | 0.040 * | 0.387 |
| Traditional daytime—Traditional nighttime | Stress | 0.661 | 0.775 | 0.934 |
| | Engagement | 0.658 | 0.827 | 0.417 |
| | Interest | 0.635 | 0.117 | 0.668 |
| | Focus | 0.171 | 0.057 | 0.559 |
| | Excitement | 0.060 | 0.355 | 0.291 |
| | Relaxation | 0.921 | 0.066 | 0.607 |

* Significant result, $p < 0.05$.

### 3.1. Al Ain City Locals

Each boxplot graph for the different participant groups is divided into three zones, each zone representing a set of statistical analyses. Figure 2 shows the differences in EEG readings derived from the brand image and foreign image scenes for the locals from Al Ain city. The zero-value line indicates no change in readings. In the left-hand and middle zones, a positive result indicates that the brand image readings were higher than the foreign image readings, and a negative result indicates that the foreign image readings were higher than the brand image readings. In the right-hand zone, a positive result indicates that the daytime brand image readings were higher than the nighttime brand image readings, while a negative result indicates that nighttime brand image readings were higher.

Boxplots in red indicate that the performance metric was statistically significant based on the *p* values.

We interpret the significant Wilcoxon signed-rank test findings for the Al Ain locals as follows. Their focus when observing the daytime brand image scene and the modern scenes was, on average, higher for the daytime brand image scene (Mdn = 65.3) than for the modern scenes (Mdn = 57.8). The Wilcoxon signed-rank test indicated that this difference was statistically significant (T = 27; Z = −2.197; $p < 0.029$). The excitement of nonlocals when observing the daytime brand image scene compared with the modern scenes was, on average, higher for the daytime brand image scene (Mdn = 65.3) than for the modern scenes (Mdn = 41.2). The Wilcoxon signed-rank test indicated that this difference was statistically significant (T = 26; Z = −2.028; $p < 0.044$). The excitement of nonlocals when observing the nighttime brand image scene compared with the modern scenes was, on average, higher for the nighttime brand image scene (Mdn = 60) than for the modern scenes (Mdn = 41.2). The *p* values from the bootstrap for the paired *t*-test were <0.026 for focus and <0.037 for excitement when observing the daytime brand image scene compared with the modern scenes.

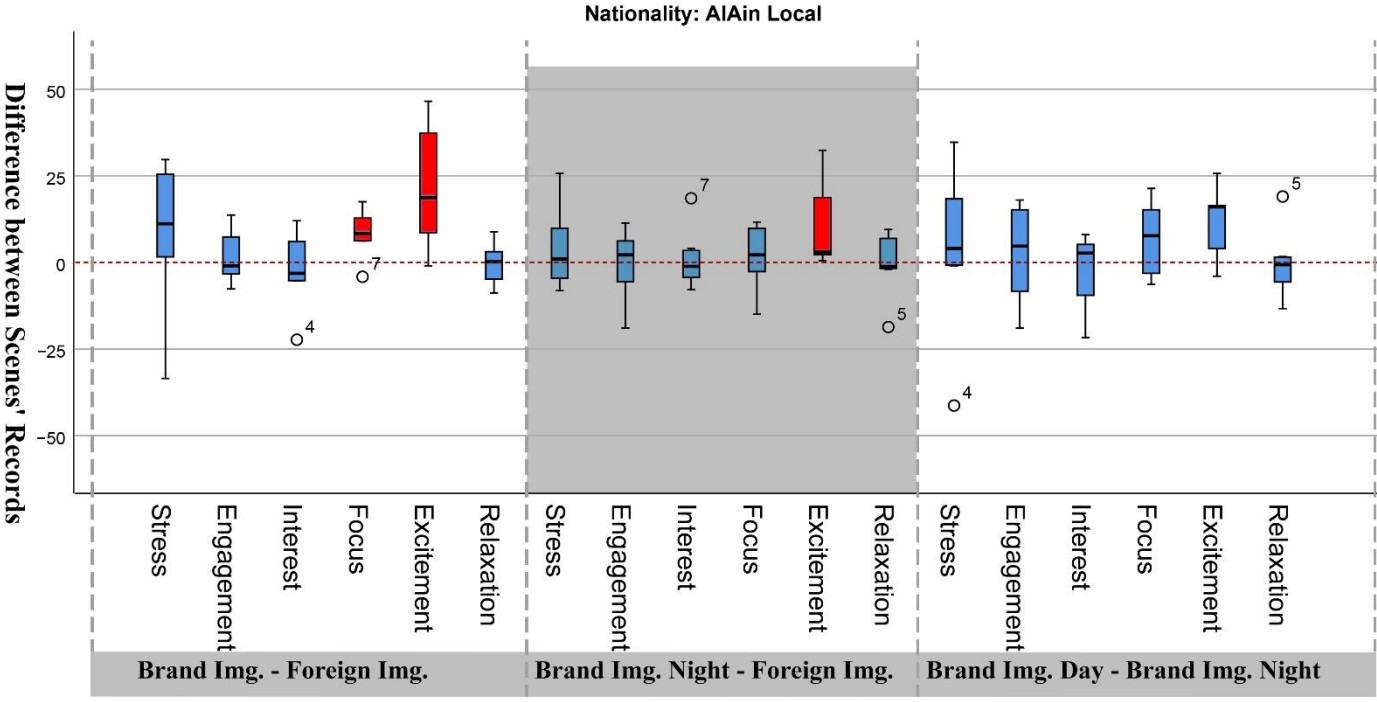

**Figure 2.** Wilcoxon analysis of EEG data from Al Ain locals.

### 3.2. Northern Emirates Locals

Figure 3 shows the differences in EEG readings between the brand image and foreign image scenes for locals from the northern emirates. The zero-value line indicates no change in readings. In the left-hand and middle zones, a positive result indicates that the brand image readings were higher than the foreign image readings, and a negative result indicates that the foreign image readings were higher than the brand image readings. In the right-hand zone, a positive result indicates that the daytime brand image readings were higher than the nighttime brand image readings, and a negative result indicates that nighttime brand image readings were higher than the daytime brand image readings. Boxplots in red indicate that the performance metric was statistically significant based on the *p* values.

### 3.3. Nonlocals

Figure 4 shows the differences in EEG readings between the brand image and modern scenes for non-UAE locals. The zero-value line indicates no change in readings. In the left-hand and middle zones, a positive result indicates that the brand image readings were higher than the foreign image readings, and a negative result indicates that the foreign image readings were higher than the brand image readings. In the right-hand zone, a positive result indicates that the daytime brand image readings were higher than the nighttime brand image readings, and a negative result indicates that nighttime brand image readings were higher than the daytime brand image readings. Boxplots in red indicate that the performance metric was statistically significant based on the *p* values.

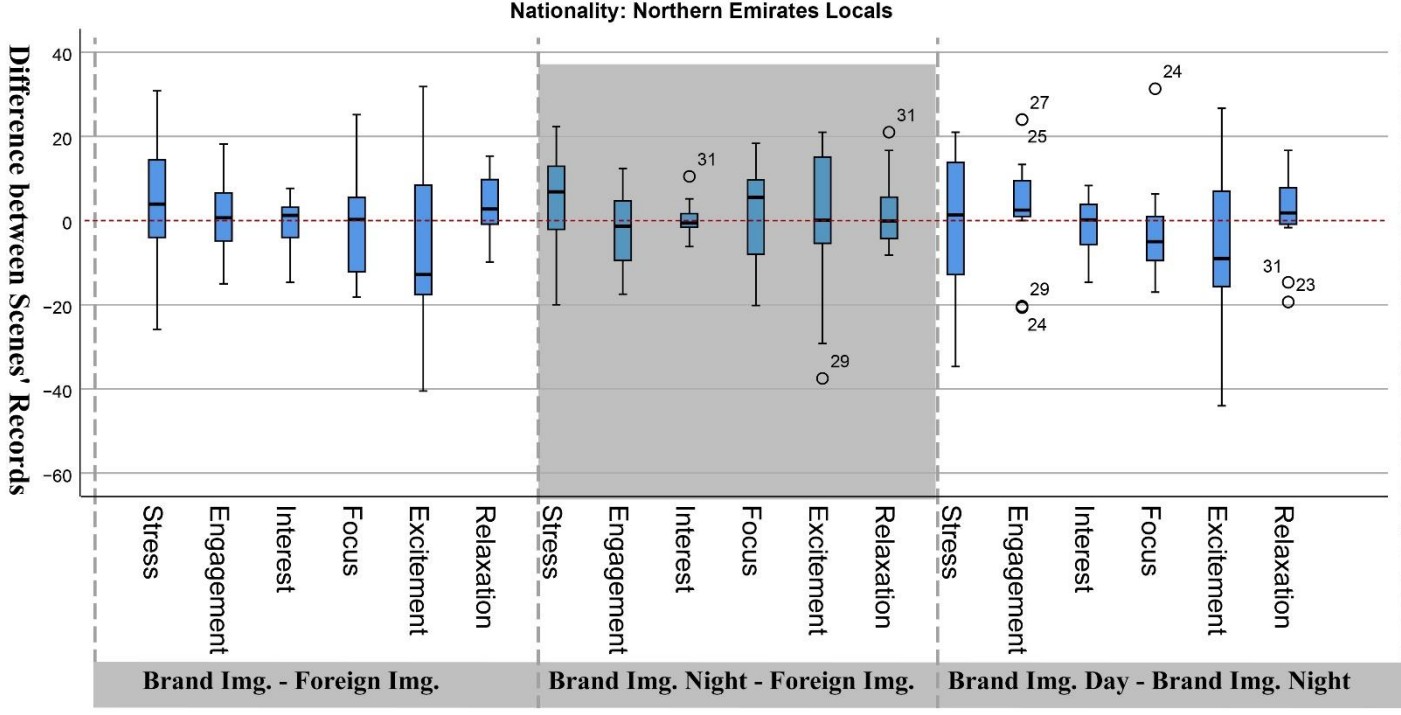

**Statistically non-significant value**

**Figure 3.** Wilcoxon analysis of EEG data from northern emirates locals.

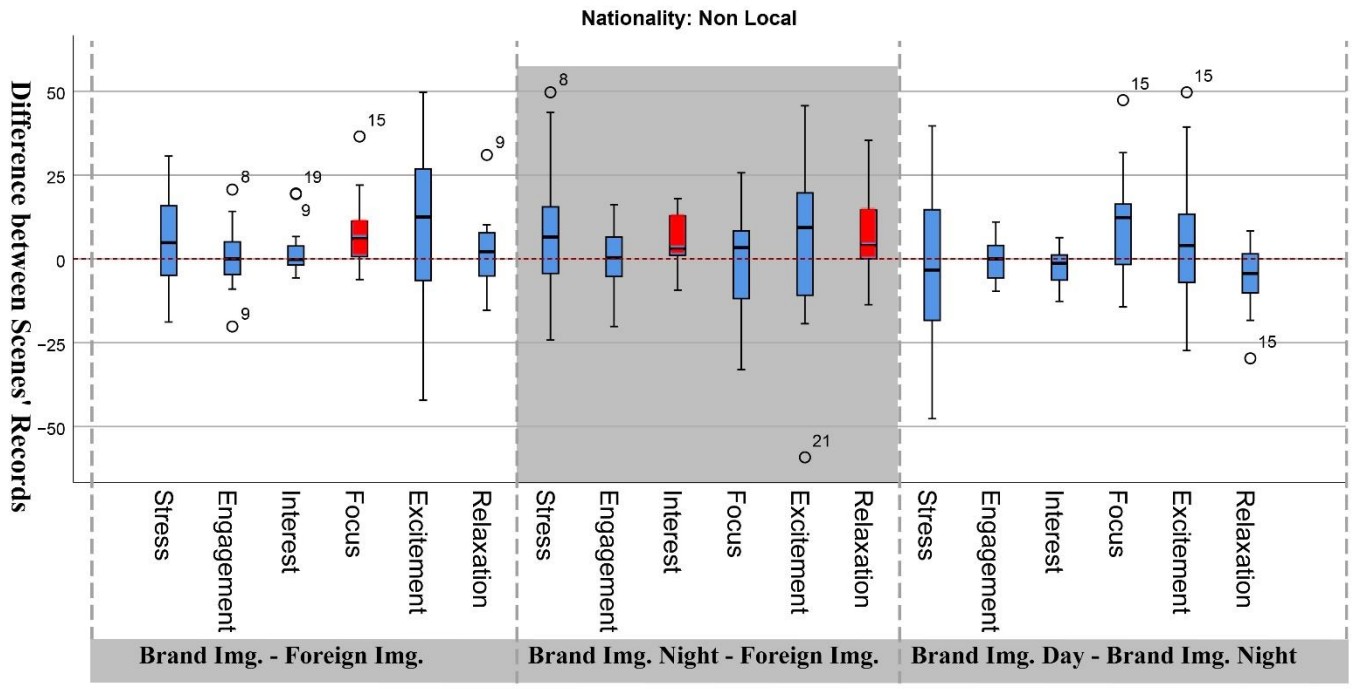

**Statistical significant value** *p* < 0.05

**Figure 4.** Wilcoxon analysis of EEG data from nonlocals.

We interpreted the significant changes among the nonlocals as follows. Their focus when observing the daytime brand image scene compared with the modern scenes was, on average, higher for the daytime brand image scene (Mdn = 62.3) than for the modern scenes (Mdn = 52.8). The Wilcoxon signed-rank test indicated that this difference was statistically

significant (T = 104; Z = −2.5; *p* < 0.013). The nonlocal participants' interest when observing the nighttime brand image scene compared with the modern scenes was, on average, higher for the nighttime brand image scene (Mdn = 65) than for the modern scenes (Mdn = 56.3). The Wilcoxon signed-rank test indicated that this difference was statistically significant (T = 106; Z = −2.613; *p* < 0.010). The nonlocal participants' relaxation when observing the nighttime brand image scene compared with the modern scenes was, on average, higher for the nighttime brand image scene (Mdn = 42) than for the modern scenes (Mdn = 29.3). The Wilcoxon signed-rank test indicated that this difference was statistically significant (T = 95; Z = −1.988; *p* < 0.048). The *p* value from the paired *t*-test was < 0.016 for focus when observing the daytime brand image scene compared with the modern scenes, and from the bootstrap for the paired t test, *p* < 0.041 for focus when observing the daytime brand image scene compared with the modern scenes. The *p* values from the bootstrap for the paired *t*-test were < 0.022 for interest when observing the nighttime brand image scene compared with the modern scenes and < 0.040 for relaxation when observing the same scenes.

*3.4. Interviews Analysis and Results*

We analyzed the interview data through a deductive content analysis approach. Initially, we read each participant's transcribed interview responses repeatedly and analyzed them separately (within-case analysis) to build a profile of each architect's prior experience in the UAE, his knowledge about traditional architecture and architecture theory, his impressions of both areas, and his interpretations of the empirical study's results. Then, for the group case analysis, we categorized the important ideas we identified, and themes were developed. We brought all vignettes from the six participants into the four themes that guided the interview questions: the architects' experience, knowledge, impressions, and interpretation. Each architect provided us with ideas and suggestions under each theme, which we discuss below. Table 7 summarizes the architects' most important interview excerpts related to the four themes.

**Table 7.** Summary of the architect interviews.

| Architectural Style Paradigms | | |
|---|---|---|
| **Theme** | **Vignette** | **Architect** |
| Experience | *"Jordanian architect with 6 years of experience in UAE"* | Arch 1 |
| | *"Egyptian architect with 5 years of experience in UAE"* | Arch 2 |
| | *"Jordanian Architect with 9 years in Abu Dhabi"* | Arch 3 |
| | *"Jordanian architect with 6 years of experience in UAE"* | Arch 4 |
| | *"Urban manager in Dubai authority for planning"* | Arch 5 |
| | *"Emirati architect with 13 years of experience"* | Arch 6 |
| Knowledge | *"Traditional building, and modernity buildings"* | Arch 1 |
| | *"Traditional architecture and modern architecture"* | Arch 2 |
| | *"This is Barjeel style which is the dominant style for the traditional architecture of UAE while the other style is modern buildings"* | Arch 3 |
| | *"I think this is a minimalism style designed based on the less is more idea"* | Arch 4 |
| | *"Traditional architecture, this is the origin of traditional architecture of Dubai"* | Arch 5 |
| | *"Traditional architecture, which is spread around in Dubai, Abu Dhabi, and Sharjah"* | Arch 6 |
| Impression | Traditional buildings: *"I feel warm and cozy. You feel like this type of architecture is close to the people close to me, as you can see the wood material and mud all the materials are close to the people"*. Modern buildings: *"comfortable and practical but it doesn't afford the intimate and warm feeling"*. | Arch 1 |
| | Traditional: *"The roughness of the surfaces gives the feeling of authenticity, which is similar to the skin of the elderly people"* Modern: *"I didn't feel that the place is inviting me to stay. It might be because of the colors"*. | Arch 2 |
| | *"daylight is more beautiful than that with artificial light"* | Arch 5 |
| | *"There is a missing link between the traditional architecture and the modern needs of modern life"*. | Arch 5, 6 |

**Table 7.** *Cont.*

| Theme | Architectural Style Paradigms | | |
|---|---|---|---|
| | Vignette | | Architect |
| interpretation | Alain locals | *"locals here are connected with this architecture because they feel it is a part of their customs and traditions . . . also because it is always within their sight specially . . . and activities are conducted in those places"* | Arch 1 |
| | | *"Hmmm I think then it is the place of being raised"* | Arch 2 |
| | | *"their attachment is something very expected and understandable"* | Arch 3 |
| | | *"Alain locals have been raised and grown in this kind of architecture"* | Arch 4 |
| | | *"locals of Alain are people of who love the quietness, love the nature, the palms, the water, oases, sand"* | Arch 6 |
| | Expats | *"same architectural elements between the traditional architecture of UAE and his country (wood, hay, mud) also the handcrafting skill . . . emotional part in the expats because of their nostalgia to their home countries . . . the uniqueness is another important issue"* | Arch 1 |
| | | *"they felt the nostalgia to their home countries"* | Arch 2 |
| | | *"there is a kind of similarity in some elements that you might see here and there"* | Arch 3 |
| | | *"they are living in Alain, so they have built a connection with the traditional architecture of that city"* | Arch 4 |
| | | *"might be because of their knowledge about the history of the UAE, they have been affected"* | Arch 6 |
| | Northern Emiratis | *"north emirates are paying much less attention in showcasing the forts . . . and it is far from the centers of the cities"* | Arch 1 |
| | | *"Hmmm I think then it is the place of being raised"* | Arch 2 |
| | | *"I felt a different mentality for them, I felt they have more open minded mentality"* | Arch 3 |
| | | *"I don't think in the mountains areas they were paying attention to the barjeels . . . since it is in mountains it is surely something different"* | Arch 4 |
| | | *"modernity is there but of course it is not comparable with Dubai or Abu Dhabi nor Sharjah"* | Arch 6 |

### 3.4.1. Experience

We interviewed three architects from Jordan who had lived in the UAE with 5–10 years of experience in the UAE (Arch 1, Arch 3, and Arch 4), two architects from the UAE with experience of more than 10 years in the UAE, and one architect from Egypt who had lived in the UAE with 5 years of experience in the UAE. All six architects were male. Two of the Jordanian architects (Arch 1 and Arch 3) had worked for northern emirate locals and Alain locals, and the third architect (Arch 4) had not worked for northern emirate locals. The Egyptian architect (Arch 2) was similar to Arch 4 in having no experience working with locals from the northern emirates, but he had worked with locals from Alain city. The local architects (Arch 5 and Arch 6) showed wide experience in working everywhere in the UAE.

### 3.4.2. Knowledge

When we asked the architects what they would call the architectural styles in the scenes we showed them and the architectural theories behind the buildings, four of the six described the buildings as traditional or modern styles. Arch 3 confidently described the Al-Fahidi district as Barjeel (the local name for a wind catcher): "This is Barjeel style which is the dominant style for the traditional architecture of UAE." Arch 4 described the city walk scene as minimalist and "based on the less is more idea," but the other five architects confidently described the city walk as modern. However, none of the architects identified the buildings in the scenes we showed them as reflecting any of the styles or movements described by the architectural theorists in Section 1.1 of this article. We found this an interesting outcome that might highlight a gap between the theory and the architects' knowledge.

### 3.4.3. Impressions

The architects' impressions about the four scenes varied significantly, with most (Arch 1, 2, 3, and 6) showing a preference for the traditional architecture (brand image). Arch 4

preferred the foreign image, and Arch 5 reported no explicit preference but rather talked about them all in detail. However, Arch 5 did prefer the daylight brand image over the nighttime scene: "The night light have been used to attract visitors, but in my opinion the daylight is more beautiful than that with artificial light. The daylight I call it purist friendly." He also stated, "There is a missing link between the traditional architecture and the modern needs of modern life," which we interpret as indicating this architect's awareness of and appreciation for the traditional architecture.

3.4.4. Interpretation

All of the architects provided us with at least partial interpretations of our empirical study findings, except Arch 5, who avoided doing so. However, Arch 1 considered the significant preference among the locals from Alain city for the tradition (brand image) scene because of attachments to their heritage and because their own architecture was all around them:

> *Alain city in the perspective of locals is the origin, even it is said that the UAE rulers are came from this city, also the tribes existing in Alain are still connected with its customs and traditions because the center of this tribes is still within this area. Therefore, locals here are connected with this architecture because they feel it is a part of their customs and traditions, and also because it is always within their sight specially that Alain contains many forts and castles that is well maintained, and many modern and traditional events and activities are conducted in those places.*

Arch 5 explained the northern emirate locals' preferences as follows:

> *while north emirates are paying much less attention in showcasing the forts even though the northern emirates have a quit good amount of forts but it is not prominent in general and it is far from the centers of the cities also the locals contact with this forts are less than Alain.*

Lastly, regarding the expat group, Arch 5 justified their attachment to the traditional architecture of Alain based on:

(1)   their nostalgia for the traditional architecture of their home countries:

> *me as a person came from Jordan or from Egypt, I used to see the modern architecture everywhere; I see it on the TV, in many areas in our countries. It is a repetitive for me. While the traditional part I feel -and this is an important point- it is connected to some extent with the emotional side which is already raging or we can say the intimate emotional part in the expats because of their nostalgia to their home countries and their nostalgia to details related to their home countries.*

(2)   the similarities between the traditional architecture of Jordan and Egypt on one side and that of the UAE on the other:

> *I think also there is an important point for any person; architect or layperson. Once he see the same architectural elements between the traditional architecture of UAE and his country (wood, hay, mud) also the handcrafting skill it keeps him feel with the connection with this architecture.*

(3)   the attractiveness and uniqueness of the brand design in the UAE:

> *Also the uniqueness is another important issue, for example if Al-Fahidi development was very common and spread in Dubai and in every street you see something similar, people won't feel this uniqueness.*

Arch 2, 3, and 4 confirmed Arch 1's interpretation that locals from Alain are attached to the traditional architecture because they see it everywhere; Arch 3 even commented that "their attachment is something very expected and understandable." Arch 2 highlighted the Alain locals' attachment to the area and its architecture because it is where they were raised and the architecture connects with their families and traditions:

*Hmmm I think then it is the place of being raised ... other portion have been raised here in UAE between his family members and experiencing the traditional activities such as grouping and gathering in the desert and similar activities which increase their attachment with the traditional architecture.*

Arch 4 shared a similar interpretation:

*I think Alain locals have been raised and grown in this kind of architecture, so they are familiar with this architecture. So I think looking at this architecture reminds them with the old days and reminds them with their ancestors' houses.*

Arch 6 highlighted a different reason for the attachment of Alain locals to the brand image: "locals of Alain are people who love the quietness, love the nature, the palms the water, oases, sand."

Furthermore, the architects' interpretations of the neutral attachment of northern Emiratis varied between confirming, wondering, and partially denying. Arch 2 initially thought age might play a role: "The age group of people might affect their attachment to the traditional architect, especially people above 50 years old." However, when we shared the age range of the group, he reverted to his explanation that they were attached to the place where they were raised. Arch 3, meanwhile, thought that northern Emiratis were simply open-minded and free from attachments to the traditional architecture: "I interacted with very limited number of locals from there and limited projects, but I felt a different mentality for them, I felt they have more open-minded mentality."

Arch 4 acknowledged his limited knowledge about the traditional architecture of the northern emirates, but he was fairly sure that the coast and desert areas had different traditional architecture from that in the mountains and that the differences could have explained the neutral attitudes among the northerners: "I don't think in the mountains areas they were paying attention to the barjeels. I don't know what they had there but since it is in mountains it is surely something different." Arch 5 did not attempt to interpret the neutral attachment of the northern Emiratis, but he did consider one possibility: "Modernity is there, but of course it is not comparable with Dubai or Abu Dhabi nor Sharjah."

Regarding the expats' responses to the study images, the architects' interpretations of their results also varied, but most considered that the traditional UAE architecture invoked nostalgia for their home countries because of similarities in elements such as materials and colors used:

*Egypt and Jordan have traditional architecture which is a kind of similar to UAE's traditional architecture in term of colors. The brownish colors in the mountains in Jordan also the temples and pyramids' colors in Egypt. Therefore, it might be that expats once they saw UAE's traditional architecture which has similar colors to their traditional ones, it might be that they felt the nostalgia to their home countries in opposite to what they feel when looking at modern architecture.* (Arch 2)

Arch 3 observed, "There is a kind of similarity in some elements that you might see here and there, the arches for example you might see them in some mosques in Jordan and Egypt also the design of the parapet." Arch 4 and Arch 6 proposed an opposite interpretation; that the expats might have developed attachments to the traditional architecture of UAE in general or Alain specifically:

*I think that this also because they are living in Alain, so they have built a connection with the traditional architecture of that city.* (Arch 4)

*I don't know exactly why they have been attached to this architecture but it might be because of their knowledge about the history of the UAE, they have been affected, most of my expats friends love to visit this area.* (Arch 6)

## 4. Discussion

The results for the group of study participants who were local to Al Ain supported Agiel's findings [27], which showed that locals of Tripoli, Ghadames, and Yefren were more

emotionally connected to their vernacular architecture. In the present study, the results provided a new sign of emotional connection with the local brand image: locals showed significant increases in focus and excitement when they were observing the UAE's coastal brand image. The results for the northern emirates group showed no statistical significance, but their neutrality is an interesting topic for future studies, especially because it could indicate a weak emotional connection between northern emirate locals and the UAE coastal brand image. That is, the northern emirate locals, coming from the mountains, might be more attached to their own mountain brand image built with stone, rather than the mud and coral that was adopted in Dubai and Abu Dhabi. However, these results could also have been attributable to the youth of the participants, who were all in their early 20s, which could have affected their knowledge about their own brand images. In either case, this group of locals could be the subject of future studies to investigate their emotional connections with the nearby brand images of architecture.

The results for the nonlocals are interesting because they show that participants from other areas have a strong emotional connection with the UAE's coastal brand image in terms of focus, interest, and relaxation. This could be attributable to the similar building materials and colors used in the participants' regional vernacular architecture and those that constitute UAE's coastal brand image; indeed, one participant stated that the buildings reminded him of the vernacular buildings in his region. However, when comparing these results with those of the Al Ain locals, the findings for the Al Ain locals show significant focus and excitement, whereas the nonlocals showed significant interest, focus, and relaxation. These findings could indicate differences in laypeople's perceptions of traditional architecture; people who are unfamiliar with the architectural style might be interested in exploring it and feel relaxed when observing it, and those who are familiar with the style might feel excited when observing their own traditional architecture.

## 5. Conclusions

The argument in this manuscript relates to the effects of architectural styles on laypeople's emotions, a topic that has had limited research. Researchers such as Jack Nasar and Arthur Stamps [30,31] conducted studies on laypeople's architectural preferences. Some years later, Agiel [27] studied the emotional connections of participants with brand image (traditional) architecture. In the present study, we studied emotional effects using a different approach from those of previous studies.

The purpose of this study was to explore the effects of architectural style on six emotions by analyzing physiological responses of the brain cortex. Although there are more than six emotions, measuring all emotions was not possible, especially given that the specific number of emotions is debatable; however, the EPOC+ tool provided the possibility of measuring six emotions based on EEG waves. The findings show that traditional architecture affords a better emotional quality for laypeople living in conservative environments. Participants from Jordan and Egypt, as well as those from Al Ain city, had better emotional connections with the UAE's coastal brand image architecture. In contrast, participants from the northern emirates were not emotionally affected by the coastal brand image architecture. These results might be helpful in developing architecture practices in the UAE and developing the UAE building code, and they could also be used as a reference for unifying the architectural style of future government, health care, and institutional buildings, especially in the areas for which we confirmed participants' emotional connections with brand image and participants. However, these recommendations should be taken more seriously once a broader study confirms the results of this study.

The limitations of this study were the number and the gender of the participants. Only 29 people participated, and all were men; women were not willing to participate because the experiment required connecting the EPOC+ sensors to the head, and because of the sensors' sensitivity to long hair, which made the signal unstable. However, the results are still useful in providing a general concept of the emotional connections of people from the UAE, Jordan, and Egypt with the UAE's coastal brand image architecture. The results also

provide additional evidence that architectural styles motivate brain electrical activity in different patterns. In some cases, the experiment was interrupted by unexpected noise from the air conditioner or doors opening and closing; however, this was addressed by ignoring the results from interrupted experiments and repeating the measurements. Some of the insignificant values might also have been a result of the low sample size.

In addition to calling for broader samples for the purpose of generalizing this study or comparing both genders' results, the results of this study open the doors to future studies in the same field and context that change one major variable, such as the architectural styles, the participants, or the methodology, to identify specific changes. For example, researchers could compare foreigners' emotional connections with UAE brand image architecture or compare the emotional connections of UAE locals on one side and local image and brand image on the other side.

**Author Contributions:** Conceptualization, A.A. (Ala' Albdour) and A.A. (Ahmed Agiel); methodology, A.A. (Ala' Albdour) and A.A. (Ahmed Agiel).; software, A.A. (Ala' Albdour) and K.G.; validation, K.G., A.A. (Ala' Albdour) and A.A. (Ahmed Agiel); formal analysis, K.G. and A.A. (Ala' Albdour); investigation, A.A. (Ala' Albdour); resources, A.A. (Ala' Albdour); data curation, A.A. (Ala' Albdour); writing—original draft preparation, A.A. (Ala' Albdour); writing—review and editing, A.A. (Ahmed Agiel), K.G.; visualization, A.A. (Ala' Albdour); supervision, A.A. (Ahmed Agiel); project administration, A.A. (Ahmed Agiel); funding acquisition, A.A. (Ahmed Agiel). All authors have read and agreed to the published version of the manuscript.

**Funding:** This research was funded by United Arab Emirates University, grant number G00003445, and the APC was funded by United Arab Emirates University.

**Data Availability Statement:** The data presented in this study are available upon reasonable request from the corresponding author.

**Acknowledgments:** We would like to express our special thanks to all the participants and architects who accepted to participate in this study, especially Odai Harahsheh, Abdelrahman Amin, Mohammed Shwayat, Omar Udellat, Ahmed Bukhash and Abulaziz Kannoun. Additionally, we appreciate the efforts of Dubai Film to make the Dubai360 application and provide 360 photos for most of the places and areas of Dubai.

**Conflicts of Interest:** The authors declare no conflict of interest.

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
