# Peer review of "Assessing the Emotional Affordance of Brand Image and Foreign Image Based on a Physiological Method Using Examples from Dubai: Exploratory Study"

_buildings, doi:10.3390/buildings12101650_

Round 1

Reviewer 1 Report

I have reviewed this article that is focused to investigate a group of participants’ unconscious emotional responses among images of traditional and modern architecture in the United Arab Emirates (UAE).

The study is complete and soundly built with a topic of undoubted interest and very current. The holistic methodology of working with people to assess their reactions and responses to spatial solicitations is slowly becoming established and this article will undoubtedly contribute to its development. The objective of evaluating, in a scientific and quantitative way, the emotion that architecture produces in human beings and, specifically, evaluating their behaviour in the face of certain spatial and environmental solicitations, is a highly topical matter and is currently being studied by Neuroarchitecture, a growing discipline since 2003.

That is why in the theoretical background some references are missing, basically, to the works of that discipline, such as the researchs of Hölscher, Büchner, Meilinger, and Strube, or the interesting investigations of Valero-Flores on the behaviour of Alzheimer's patients in certain domestic spaces. The contributions of the previous researchers, among others, make the next affirmation of the authors of the paper, in line 65: “the unconscious interpretation HAS NEVER BEEN previously explored", very risky. I miss some dimensions that are linked to Neuroarchitecture.

Anyway, the authors explain their procedures in detail and use their own methodology, but it would be advisable to compare it with similar methodologies that are currently being applied by other scholars from Neuroarchitecture or Ambiental Psychology as Kaplan and Kaplan. Progress on results would perhaps be faster and more aligned with similar research being done internationally on this same subject.

As the authors explain, a limitation of this study has been the number of participants and their gender. This is why, together with other research groups and researchers from other countries, the method set out in this article can be applied in the future to a larger sample and therefore the results can be more widely transferred to the society end the impact of this kind of research it will be higher.

Reviewer 2 Report

buildings-1944816

The authors for the entitled manuscript “Assessing the Emotional Affordance of Brand Image and Foreign Image Based on Physiological Method Using Examples 3 from Dubai” explored the effects of architectural style on six emotions by analysing physiological responses of the brain cortex. The article is well articulated however, I have the following concerns:

1.     The authors did not mention the gender of the participants until the end of the article, in conclusion section. Nor did the authors describe how this compares to the gender ratio in the country.

2.     All the participants in the survey were males aged between 18 and 45. Please check the abstract showing (19-45). This element is not discussed, nor is it compared with the average age of the population.

3.     For Figure 1, please provide the year in which each of the building’s construction was completed.

4.     Also, the six architects participating in this study were males. What was the reason for this selection?

5.     This article was missing discussion on sustainability factors in the design.

6.     Whilst the methodology was sound, the sampling was at issue. The authors mentioned why females did not participate, but that does limit the applicability of the results.

7.     I would like to see more diversity in this study.

To recommend this article for accept, this lack of representation needs to be addressed. At this point it reads more like a pilot study, especially given the 29 participants and the lack of gender balance (local and nonlocal), so this should be reflected in the title.

Round 2

Reviewer 2 Report

I am happy with the manuscript revision and authors’ response.